# Factors affecting utilization of outpatient healthcare services among the elderly population in Butiama and Musoma districts, Tanzania: A community-based cross-sectional study

Magnus Michael Sichalwe[1,2]☯, Chrisostom Charles Mwesiga[3,4]☯ *, Anna Tengia Kessy[5], Manas Ranjan Behera[6]

1 Family Welfare, Primary Health and Preventive Services Organisation, Morogoro, Tanzania, 2 Department of Community Health, Butiama District Hospital, Butiama, Tanzania, 3 Muhimbili University of Health and Allied Sciences (MUHAS), Dar es Salaam, Tanzania, 4 African Medical and Research Foundation, Dar es Salaam, Tanzania, 5 Muhimbili University of Health and Allied Sciences, School of Public Health and Social Sciences, Dar es Salaam, Tanzania, 6 Kalinga Institute of Industrial Technology, Kalinga Institute of Industrial Technology Deemed to Be University, School of Public Health, Odisha, India

☯ These authors contributed equally to this work.
* chrissmwec@gmail.com

**Data Availability Statement:** We have uploaded the minimal anonymized data set necessary to

## Abstract

### Background

In Tanzania and Sub-Saharan Africa, the elderly population has grown significantly due to improved quality of life, subsequently leading to prolonged life expectancy. Despite global development initiatives, elders still face insufficient care. Through a community-based investigation, this study assessed outpatient department (OPD) healthcare utilization and its determinants among the elderly in Butiama and Musoma districts, Tanzania.

### Methodology

This study involved 415 elderly individuals aged 60 or older in Tanzania's Butiama and Musoma districts. Structured questionnaires were used to gather data, and the results were analyzed using SPSS 22. Univariate analysis utilized descriptive statistics, bivariate analysis involved cross-tabulation data, and multivariate logistic regression identified factors influencing OPD service utilization.

### Results

Approximately 43.4% of participants used OPD services in the past year. Divorced or separated individuals were over two times more likely to utilize OPD services compared to single participants. This association was statistically significant (OR 1.958; 95% CI 1.001–3.829; p = 0.05). About 74.5% of surveyed elders held a positive perception of OPD utilization. Although not statistically significant (p>0.05), individuals with a positive perception had 1.167 times higher odds of using OPD services (95% CI 0.746–1.826).

replicate our study findings as Supporting Information files.

**Funding:** The author(s) received no specific funding for this work.

**Competing interests:** The authors have declared that no competing interests exist.

## Conclusion

This study highlights a low overall utilization rate of OPD healthcare services among the elderly. Elderly individuals aged 80 years or older, along with widowed or divorced individuals, encounter specific barriers when accessing healthcare services. Positive perceptions play a crucial role in influencing healthcare utilization. It is essential to proactively offer tailored support and conduct further research, specifically addressing the distinct needs of divorced and widowed individuals when seeking healthcare services.

## Introduction

In low- and middle-income countries, the World Health Organization (WHO) classifies individuals above the age of 60 as "elderly" [1]. By 2030, one in six people globally will be 60 or older. By 2050, there will be approximately 2.1 billion more people aged 60 and above on the planet [2]. Specifically, the global population aged 80 and above is projected to triple, reaching 426 million by 2050. Tanzania reflects this demographic shift, expecting its elderly population to nearly triple from 2.95 million in 2020 to 8.39 million by 2050 [2,3]. These demographic changes have profound implications for healthcare systems and public policy, necessitating a re-evaluation of elderly healthcare and support in both global and local contexts.

Longer life expectancies yield societal benefits such as enhanced workforce wisdom, intergenerational knowledge exchange, ongoing personal development, diverse innovation, and strengthened family ties. However, the challenges of an ageing population, including increased healthcare burden, declining productivity, strained social security systems, potential resource inequities, and higher demand for long-term care, threaten productivity, economic growth, and social security sustainability [4]. Offering suitable care for the elderly poses substantial challenges as they have greater healthcare needs due to declining physical function and increased illness susceptibility [5,6]. This growing demand for healthcare services places substantial pressure on healthcare systems, necessitating the development of comprehensive and sustainable strategies to meet the unique needs of the elderly population.

The Anderson healthcare utilization model [7] illuminates challenges in the utilization of OPD services by the elderly, reflecting a global concern. Various countries, including Australia and the United States, face obstacles such as provider shortages, limited services, and inadequate facilities, affecting healthcare access [8,9]. Given that many Sub-Saharan African nations, including Tanzania, have low to moderate-income levels, similar issues likely influence older populations. Difficulty in accessing healthcare services leads to unmet healthcare needs, missed care, and suboptimal management of chronic illnesses, ultimately resulting in increased elderly visits to emergency departments [10,11].

In Tanzania, the elderly make up 5.4% of the population, with 7% of them unemployed, and 64% still engaged in informal work [12]. Additionally, the uneven distribution of medical professionals, with most in metropolitan areas where 74% of the population resides in rural areas [13], further impacts elderly healthcare utilization.

Despite government initiatives like the Community Health Fund (CHF), prioritizing elderly individuals for non-emergency care, social welfare programs with treatment waivers, and referral procedures, challenges in healthcare utilization among the elderly persist due to other socioeconomic and demographic factors [12].

Elderly people in rural Tanzania often have low education, are unemployed, lack insurance, and rely on family and neighbours for support [14,15]. This dependency can result in restricted access to healthcare and adequate nutrition for the elderly, especially when

households face financial challenges in supporting them. Also, multiple studies, including those by Frumence et al. and Tungu et al., underscore the urgent need for prioritizing research on elderly healthcare within community settings in African countries, including Tanzania [12,16]. This study centred on improving healthcare service utilization among the elderly in the Butiama and Musoma District Councils of Tanzania, with a focus on OPD services.

## Materials and methods

### Study design

The design of the study was a community-based analytical, cross-sectional study.

### Study setting and participants

The study took place in the Butiama and Musoma districts within Tanzania's Mara Region, predominantly rural areas known for activities such as agriculture, livestock keeping, fishing, and mining. The population of the Mara region was reported to be 2,372,015, with a gender distribution of 48% male and 52% female [17]. Approximately 5.6% of the region's population comprises elderly individuals [12]. Butiama district encompasses 18 administrative wards and 59 villages, hosting 49 healthcare facilities, including one hospital, four health centers, and 44 dispensaries. Meanwhile, Musoma municipal consists of 16 wards and 47 health facilities, comprising two hospitals, a health center, and 44 dispensaries.

### Study population

The study included participants aged 60 years and older residing in the Butiama district and Musoma municipal council, both located in the Mara region.

### Determination of the sample size

The sample size for the study was determined by employing Fischer's equation [18]. Due to the time lapse from the previous study [12] to the present, the prevalence for this study was considered as 50% (P = 50% = 0.5). Fischer's formula used was $N = \frac{Z^2 \times PQ}{D^2}$

Where N = Sample size
P = Prevalence, 50%
Z = Alpha risk expressed in Z score, 1.96
D = Absolute precision, 5%
Q = 1-P
Substituting the above formula,

$$N = \frac{1.96^2 \times 0.5 \times 0.5}{0.05^2}$$

The minimum sample size of 384 elderly individuals was determined based on statistical considerations to ensure adequate power for the study. Expecting 422 elders accounted for a 10% non-response rate, ensuring the study maintains a robust sample size and maintains statistical power, safeguarding against potential dropout while meeting the minimum sample size requirement.

### Sampling technique

Butiama district and Musoma municipal council were chosen conveniently for the assessment of OPD healthcare service utilization among older individuals. Butiama district and Musoma

municipal council were selected conveniently to study OPD healthcare service utilization among older individuals because of their accessibility by considering financial constraints, transportation convenience, cooperation from local authorities, and representative status within the rural Mara region. Researcher familiarity and existing connections also aided in streamlining data collection and logistics.The study employed a multistage sampling technique with five stages, as outlined in Fig 1.

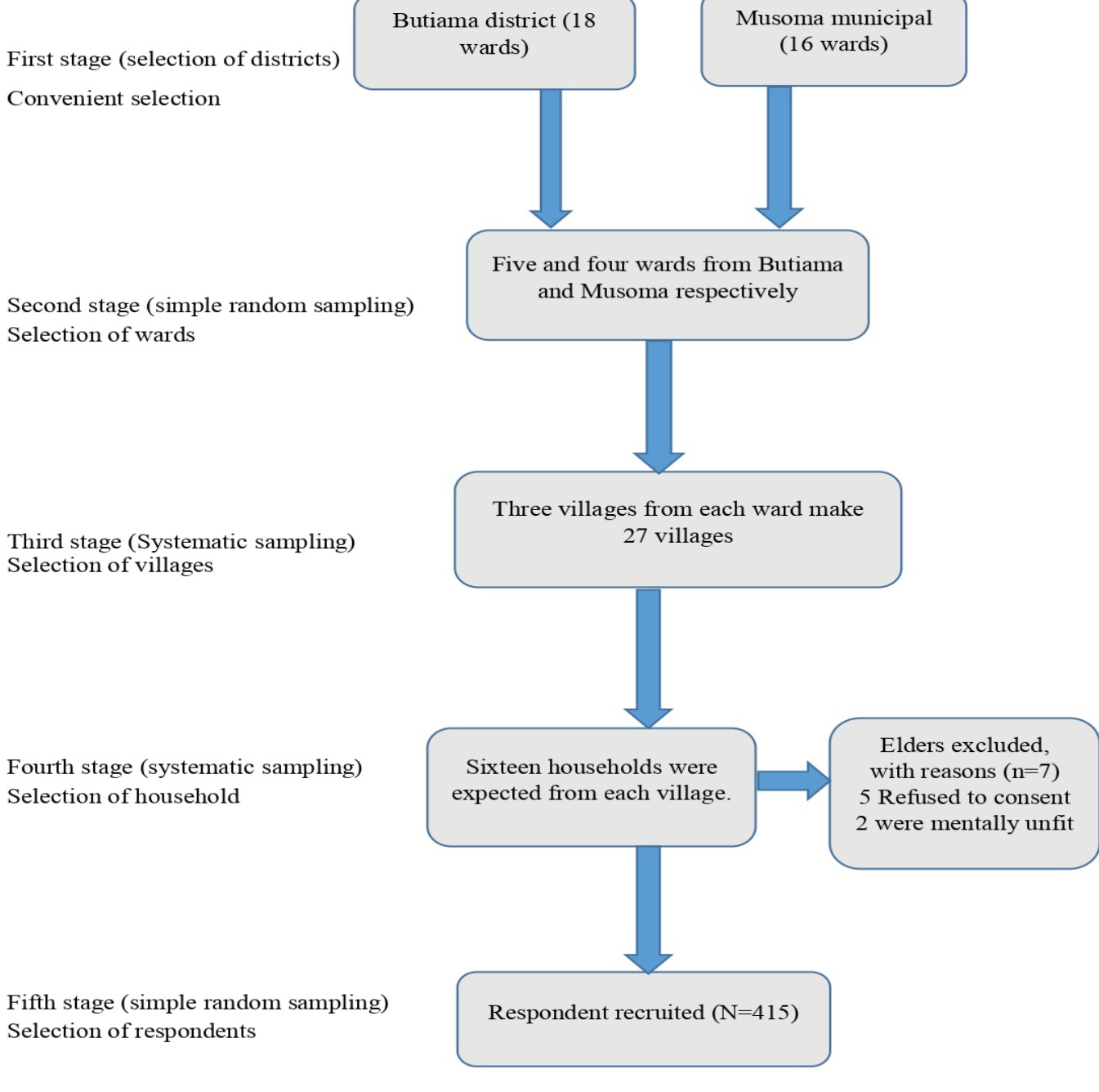

**Fig 1. Illustration of multistage sampling flow.**

### Inclusion criteria

The study included elderly individuals aged 60 years and above who lived in Musoma municipality and Butiama district and consented to participate in the study.

### Exclusion criteria

This study excluded older individuals who were severely ill, mentally incapacitated, or unwilling to give consent to participate.

### Study variables

**Dependent variables.** In this study, the utilization of OPD healthcare services served as the dependent variable. To evaluate this utilization among the elderly, the study assessed whether individuals aged 60 and above had sought healthcare services at OPD when they experienced illness in the past 12 months.

**Independent variables.** The study regarded socio-economic and demographic factors, as well as the perception of OPD healthcare utilization, as independent variables. These variables encompassed the age, educational background, gender, income, wealth index, and their perception regarding the utilization of OPD healthcare services.

Assessing wealth quantities in participants' residences defined wealth index. Participants were categorized into different economic status codes based on the percentage of household possessions, including radio, TV, mobile phone, computer, refrigerator, bicycle, animal-drawn cart, motorcycle/tricycle/scooter, car and none or only radio [19].

### Data collection methods

Three trained research assistants collected data for this study, and they were proficient in the local languages. The research instrument was pretested with 40 participants outside the study area to identify necessary modifications. The questionnaire was digitalized using the Kobo toolbox for efficient data collection, which occurred in July and August 2023.

### Data analysis

Data was entered and cleaned using SPSS version 22.0. Descriptive statistics were used to analyze socio-demographic characteristics. To calculate the weighted household wealth index, households were scored based on their possession of consumer goods or durable assets. These scores were determined through principal component analysis (PCA) in SPSS, which reduced the variables into components using loading factors as weights, ultimately yielding the weighted wealth index. Following this, households were categorized into four equal wealth quartiles, and the scores were presented as lowest, Second, Middle, and Highest wealth index.

The Pearson chi-square test identified variables with a P-value below 0.25 for potential associations [20]. These variables were considered for multinomial logistic regression. Multivariate logistic regression was used to investigate the association between dependent and independent variables, considering P values of 0.05 or below as statistically significant. Odds ratios (OR) with 95% confidence intervals measured the strength of associations.

### Ethical consideration

The study received ethical approval from the Institutional Review Board (IRB) of Muhimbili University of Health and Allied Sciences (MUHAS), identified by reference number DA.282/298//01.C/1774. Additionally, approvals were obtained from district and municipal medical officers before commencing the research in their respective jurisdictions. Courtesy visits were

made to ward and village executive officers prior to initiating the study in their designated areas. Written consent forms were provided to participants, and verbal explanations were offered to those who were unable to read or write, ensuring comprehensive understanding.

## Results and discussion

### Results

**Sociodemographic information of participants.** Table 1 summarizes the characteristics of 415 elderly participants in the study. Seven elders were unable to participate: five rejected consent and two were mentally unfit. Data were obtained from 415 instead of 422, resulting in a response rate of 98.3%. Of the 415 participants, 171 (41.2%) were male and 244 (58.8%) were female. The mean age was 69.06 ± 6.995 years, and 271(65.3%) of the sample were aged ≤70 years. Most elders were widowed 145 (34.9%) and completed primary school education 283 (68.2%). Most households 282(68.0%) had more than three family members. Most of the study participants, 269(64.8%), were engaged in unskilled labour. Most of the study participants, 254 (61.2%) had monthly income of less than 100,000 Tzs (40 USD). About three-quarters of respondents, 312(75.2%) fell within the middle range of the wealth index. In terms of lifestyle choices, 107(25.8%) of the elders reported consuming alcohol, and 44(10.6%) reported smoking cigarettes or using other tobacco products. Additionally, only about one-third 134(32.3%) of the respondents reported engaging in physical activities at the time of the study.

**Accessing OPD for care in the past 12 months.** Of the surveyed elderly individuals, 180 (43.4%) reported having visited the OPD within the past year. Among those who had visited the OPD, approximately 124(68.9%) utilized public healthcare facilities. Regarding the frequency of OPD visits in the past 12 months, 67 (37.3%) mentioned visiting OPD more than twice within the year. A total of Number 309(74.5%) of elderly individuals expressed favourable attitudes towards the utilization of OPD services. About 144(34.7%) reported unmet health needs attributed to various factors, with financial constraints being the primary concern for 326(78.6%) of all individuals.

**Factors associated with OPD visitation.** Table 2 indicates that participants aged above 80 years (56.1%) visited OPD more frequently than those aged 80 years and below. Participants with a middle wealth index (72.8%) visited OPD more often than those with lower or higher wealth indices did.

Table 3 presents the multivariate logistic regression results for the socio-economic, demographic, and other risk factors affecting OPD visitation among the elderly in the Mara Region over the past 12 months. Elders aged over 80 years were 62% less likely to visit OPD than those under 70 years, with statistical significance (OR 0.379, 95% CI 0.179–0.803, p-value = 0.011). Divorced or separated elders were almost twice as likely to visit OPD compared to single individuals, a statistically significant association (OR 1.958, 95% CI 1.001–3.829, p-value = 0.05). Widowed participants were also almost twice as likely to visit OPD compared to those who were single, with a statistically significant adjusted odds ratio of 1.985 and CI (1.024–3.849), and a p-value of 0.042.

**Perception on utilization of outpatient healthcare services.** In Table 4, it was found that 74.5% of respondents had a positive perception towards utilizing healthcare services, indicating a generally favorable attitude among the elderly towards seeking medical care.

Among the 415 participants surveyed, 269 (64.8%) strongly agreed on the importance of healthcare services, while 391 (94.2%) expressed willingness to use outpatient department (OPD) services. However, a significant portion, 354 (85.3%), considered healthcare costs to be high, and 262 (63.1%) admitted to harboring a fear of being diagnosed with a disease.

**Table 1. Social demographic characteristics of the participants (n = 415).**

| VARIABLES | N | % |
|---|---:|---|
| Gender of the elder | | |
| Male | 171 | (41.2) |
| Female | 244 | (58.8) |
| Age group (years) | | |
| ≤70 | 271 | (65.3) |
| 71–80 | 103 | (24.8) |
| ≥81 | 41 | (9.9) |
| Elder's education level | | |
| Cannot read or write | 34 | (8.2) |
| Primary level | 283 | (68.2) |
| Secondary level | 73 | (17.6) |
| College and above | 25 | (6.0) |
| Elder's Marital status status | | |
| Single | 61 | (14.7) |
| Married/Cohabiting | 107 | (25.8) |
| Divorced/Separated | 102 | (24.6) |
| Widowed | 145 | (34.9) |
| Source of income | | |
| Employment | 14 | (3.4) |
| Agriculture | 132 | (31.8) |
| unskilled labourer | 269 | (64.8) |
| Monthly income of household (Tshs) | | |
| <100 000 | 254 | (61.2) |
| 100000–500000 | 115 | (27.7) |
| >500000 | 46 | (11.1) |
| Household size | | |
| 1–2 | 48 | (11.6) |
| 3 | 85 | (20.5) |
| >3 | 282 | (68.0) |
| Alcohol consumption status | | |
| Yes | 107 | (25.8) |
| No | 308 | (74.2) |
| Cigarette or use of any tobacco products | | |
| Yes | 44 | (10.6) |
| No | 337 | (81.2) |
| Quitted | 34 | (8.2) |
| Status of physical exercises | | |
| Yes | 134 | (32.3) |
| No | 281 | (67.7) |
| Wealth Index | | |
| Lowest | 26 | (6.3) |
| Second | 69 | (16.6) |
| Middle | 312 | (75.2) |
| Highest | 8 | (1.9) |

**Table 2. Bivariate analysis of socioeconomic factors and OPD visitation in the past 12 months among the elderly in Butiama and Musoma districts.**

| Variables | Visiting OPD N (%) | Not Visiting OPD N (%) | P value |
|---|---|---|---|
| Gender | | | |
| Male | 67(39.2) | 104(60.8) | |
| Female | 113(46.3) | 131(53.7) | .149* |
| Age | | | |
| ≤70 | 108(39.9) | 163(60.1) | |
| 71–80 | 49(47.6) | 54(52.4) | |
| ≥81 | 23(56.1) | 18(43.9) | 0.09* |
| Elder's education level | | | |
| Cannot read or write | 13(38.2) | 21(61.8) | |
| Primary level | 123(43.5) | 160(56.5) | |
| Secondary level | 34(46.6) | 39(53.4) | |
| College and above | 10(40.0) | 15(60.0) | .853 |
| Elder's civil status | | | |
| Single | 31(50.8) | 30(49.2) | |
| Married/Cohabiting | 49(45.8) | 58(54.2) | |
| Divorced/Separated | 37(36.3) | 65(63.7) | |
| Widow | 63(43.4) | 82(56.6) | 0.193* |
| Source of income | | | |
| Employment | 10(71.4) | 4(28.6) | |
| Agriculture | 56(42.4) | 76(57.6) | |
| unskilled labourer | 114(42.4) | 155(57.6) | 0.717 |
| Monthly income of the Household | | | |
| <100 000 | 107(42.1) | 147(57.9) | |
| 100000–500000 | 56(48.7) | 59(51.3) | |
| >500000 | 17(37.0) | 29(63.0) | 0.323 |
| Number of family members in the household | | | |
| 1–2 | 18(37.5) | 30(62.5) | |
| 3 | 31(36.5) | 54(63.5) | |
| >3 | 131(46.5) | 151(53.5) | 0.182* |
| Alcohol consumption status | | | |
| Yes | 40(37.4) | 67(62.6) | |
| No | 140(45.5) | 168(54.5) | .147* |
| Cigarette or use of any tobacco products | | | |
| Yes | 13(29.5) | 31(70.5) | |
| No | 150(44.5) | 187(55.5) | |
| Quitted | 17(50.0) | 17(50.0) | .122* |
| Status of physical exercises | | | |
| Yes | 62(46.3) | 72(53.7) | |
| No | 118(42.0) | 163(58.0) | .411 |
| Wealth Index | | | |
| Lowest | 14(53.8) | 12(46.2) | |
| Second | 28(40.6) | 41(59.4) | |
| Middle | 131(42.0) | 181(58.0) | |
| Highest | 7(87.5) | 1(12.5) | .047* |

A star (*) denotes values with a P-value below 0.25, suggesting potential positive relationships worthy of consideration in multivariate analysis.

**Table 3. Multivariate analysis of factors influencing OPD visits among the elderly in Butiama and Musoma districts.**

| Variables | 95% C.I. for EXP(B) | | P value |
|---|---|---|---|
| | Lower | -Upper | |
| Gender | | | |
| Male | Reference | | 1 |
| Female | .870 (.557 | -1.358) | .540 |
| Age | | | |
| <70 | Reference | | |
| 71–80 | .660 (.402 | -1.087) | .102 |
| >81 | .379 (.179 | -.803) | .011* |
| Marital status | | | |
| Single | Reference | | 1 |
| Married/Cohabiting | 1.338 (.692 | -2.588) | .386 |
| Divorced/Separated | 1.958 (1.001 | -3.829) | .050* |
| Widow | 1.985 (1.024 | -3.849) | .042* |
| Number family members | | | |
| 1–2 | Reference | | 1 |
| 3 | .923 (.409 | -2.086) | .848 |
| >3 | .653 (.326 | -1.311) | .231 |
| Alcohol consumption | | | |
| Yes | Reference | | |
| No | .736 (.430 | -1.260) | .263 |
| Cigarette smoking | | | |
| Yes | Reference | | 1 |
| No | .769 (.364 | -1.624) | .491 |
| Quitted | .521 (.190 | -1.426) | .205 |
| Wealth Index | | | |
| Lowest | Reference | | 1 |
| Second | 1.481 (.565 | -3.884) | .425 |
| Middle | 1.500 (.638 | -3.530) | .353 |
| Highest | .127 (.013 | -1.252) | .077 |

A star (*) signifies statistically significant values, where the P-value is less than 0.05 and the odds ratio (OR) is greater than 1.

Interestingly, 185 (44.6%) participants did not believe that ageing increased their susceptibility to disease. Moreover, 214 (51.6%) perceived healthcare providers as potentially rude. Additionally, a notable 231 (55.6%) participants believed that private healthcare services were superior to public facilities. These findings are visually represented in Figs 2–7. providing a graphical depiction of the perceptions of healthcare utilization among the elderly participants.

While those with a favourable perception were more inclined to utilize OPD services (1.167 times higher), the absence of statistical significance and the broad confidence interval (0.746 to 1.826) necessitate additional investigation.

## Discussion

This study assessed the prevalence of outpatient healthcare service utilization among the elderly population residing in two districts of the Mara region (Butiama and Musoma districts) while also identifying the factors that may be associated with such utilization. The study results

**Table 4. Perception of elderly people towards OPD healthcare service utilization.**

| Item | Strong Agree | Agree | Neutral | Disagree | Strong disagree |
|---|---|---|---|---|---|
| 1. Need for utilizing healthcare services | 269(64.8)* | 136(32.8)* | 6(1.4) | 4(1.0) | 0(.0) |
| 2. Willing to utilize OPD healthcare services | 249(60.0)* | 142(34.2)* | 0(.0) | 24(5.8) | 0(.0) |
| 3. Willing to be screened for different diseases without having any sign or symptom | 147(35.4) | 203(48.9) | 21(5.1) | 36(8.7) | 8(1.9) |
| 4. Fear of the procedures used in the provision of healthcare services | 45(10.8) | 75(18.1) | 28(6.7) | 245(59.0) | 22(5.3) |
| 5. Feeling shy to expose private parts during the procedure to young or opposite-sex healthcare service providers | 61(14.7) | 120(28.9) | 39(9.4) | 176(42.4) | 19(4.6) |
| 6. Fear of the pain/discomfort during the procedures of healthcare services | 74(17.8) | 109(26.3) | 24(5.8) | 186(44.8) | 22(5.3) |
| 7. Fear of being diagnosed with the disease | 130(31.3)* | 132(31.8)* | 46(11.1) | 96(23.1) | 11(2.7) |
| 8. The available healthcare services are expensive | 141(34.0)* | 213(51.3)* | 32(7.7) | 26(6.3) | 3(.7) |
| 9. Susceptible to develop different diseases due to ageing processes | 44(10.6) | 82(19.8) | 88(21.2) | 185(44.6)* | 16(3.9) |
| 10. Healthcare providers are rude when providing care to patients | 214(51.6)* | 157(37.8)* | 19(4.6) | 21(5.1) | 4(1.0) |
| 11. Early utilization of healthcare services may be beneficial to health | 126(30.4) | 110(26.5) | 35(8.4) | 138(33.3) | 6(1.4) |
| 12. Private hospitals have good health services compared to government-owned hospitals | 98(23.6)* | 133(32.0)* | 56(13.5) | 125(30.1) | 3(.7) |
| 13. Waiting to see a doctor is inconvenient | 35(8.4) | 58(14.0) | 118(28.4) | 176(42.4) | 28(6.7) |

Highlighted with a star (*), these values hold particular significance for a deeper comprehension of elderly individuals' perceptions.

revealed that only 43.4% of the participants used the outpatient healthcare services available. In the Butiama area, for instance, there is only one healthcare facility for every two villages, which can be seen as a limitation regarding accessibility to healthcare services. Of the 47 health facilities available in the area, 44 are dispensaries, which can accommodate at least 5000–10000 individuals. Consequently, the proportion of elderly people utilizing OPD services in Butiama could be much higher and denote underutilization. Interestingly, this outcome aligns with the findings of a similar study conducted by Cheboi, 2017 in Kibera, Nairobi, Kenya, where there was a 40.4% utilize of healthcare services [21]. This consistent pattern of underutilization across different regions underscores the importance of addressing these challenges to ensure that the elderly population receives the healthcare they require.

Following multivariate analysis, our study found that individuals aged 80 and above were significantly less likely to use outpatient healthcare services compared to those under 70, while divorced/separated and widowed individuals showed increased utilization. Those aged 80 years or older experienced a significant 62% reduction in OPD healthcare service utilization. It is important to note that the mean age of the participants in this study was 69±9 years, indicating that most of the population studied was in the age of utilizing these services. Nevertheless, the decline in utilization among the elderly necessitates a substantiating study in our local population, as studies in the literature have shown a decrease in outpatient clinic utilization with advancing age [22,23]. These results emphasize the significance of considering age subgroups in healthcare planning for the elderly. They suggest potential compromised health conditions and an ongoing need for medical care that may surpass the capacity of standard healthcare services [22]. Thus, this knowledge gap requires further research to understand our specific context better.

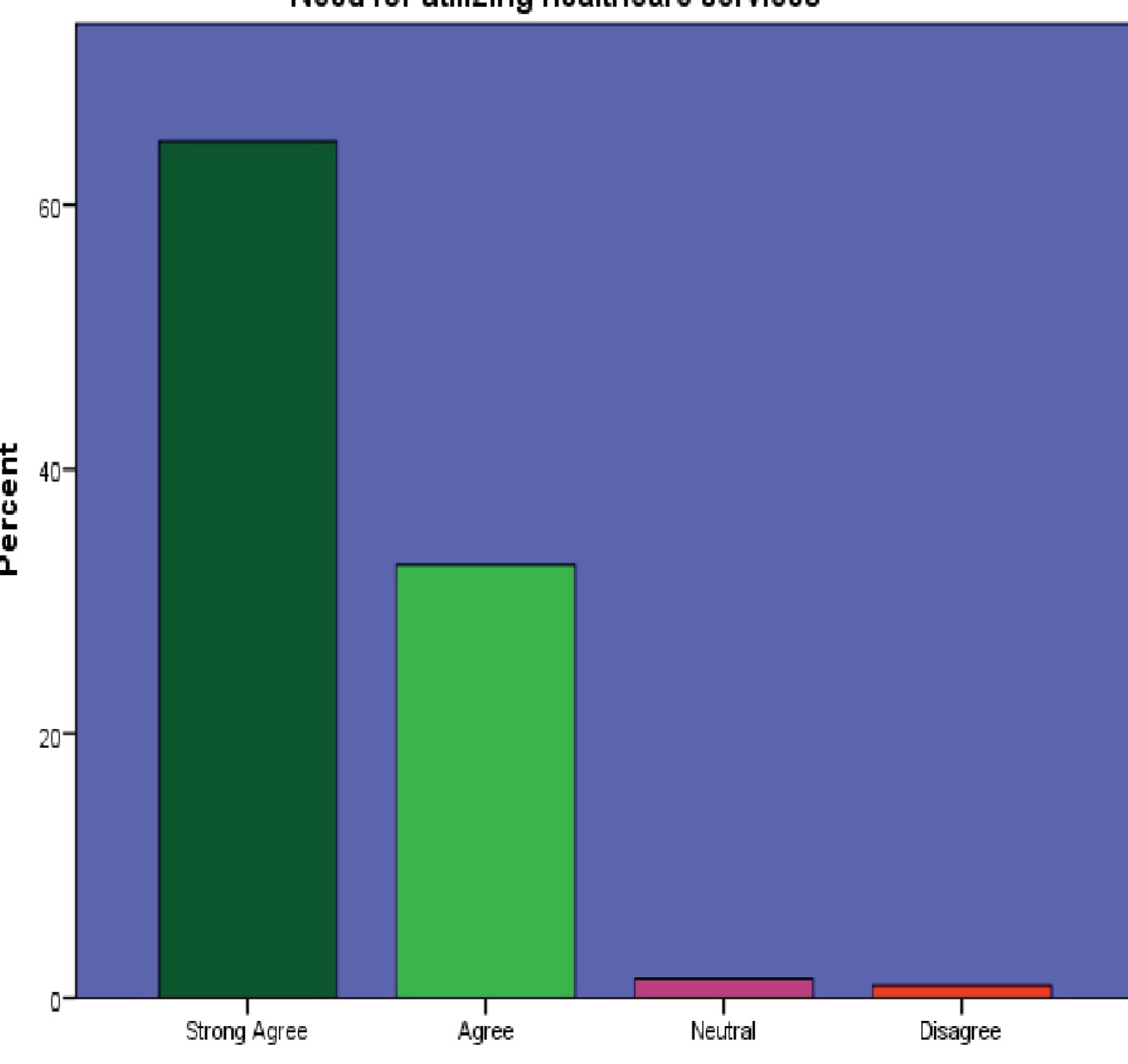

**Fig 2. Graphical representation of participants' perception of the need of utilizing healthcare services.**

Participants who reported being widowed during the study were found to have increased utilization of outpatient healthcare services twice compared to single participants. Traditionally, older widows lack family networks and spousal support to maintain better health. This situation usually worsens health status, and therefore, this group is more likely to use healthcare services than those whose spouses are still alive. A longitudinal analysis study across Europe confirmed that being widowed hurts well-being and mental health and causes a greater consumption of long-term care services [24]. The effects are more potent in the short term. In the long term, the adaptive effects smooth the impact on well-being and long-term care utilization

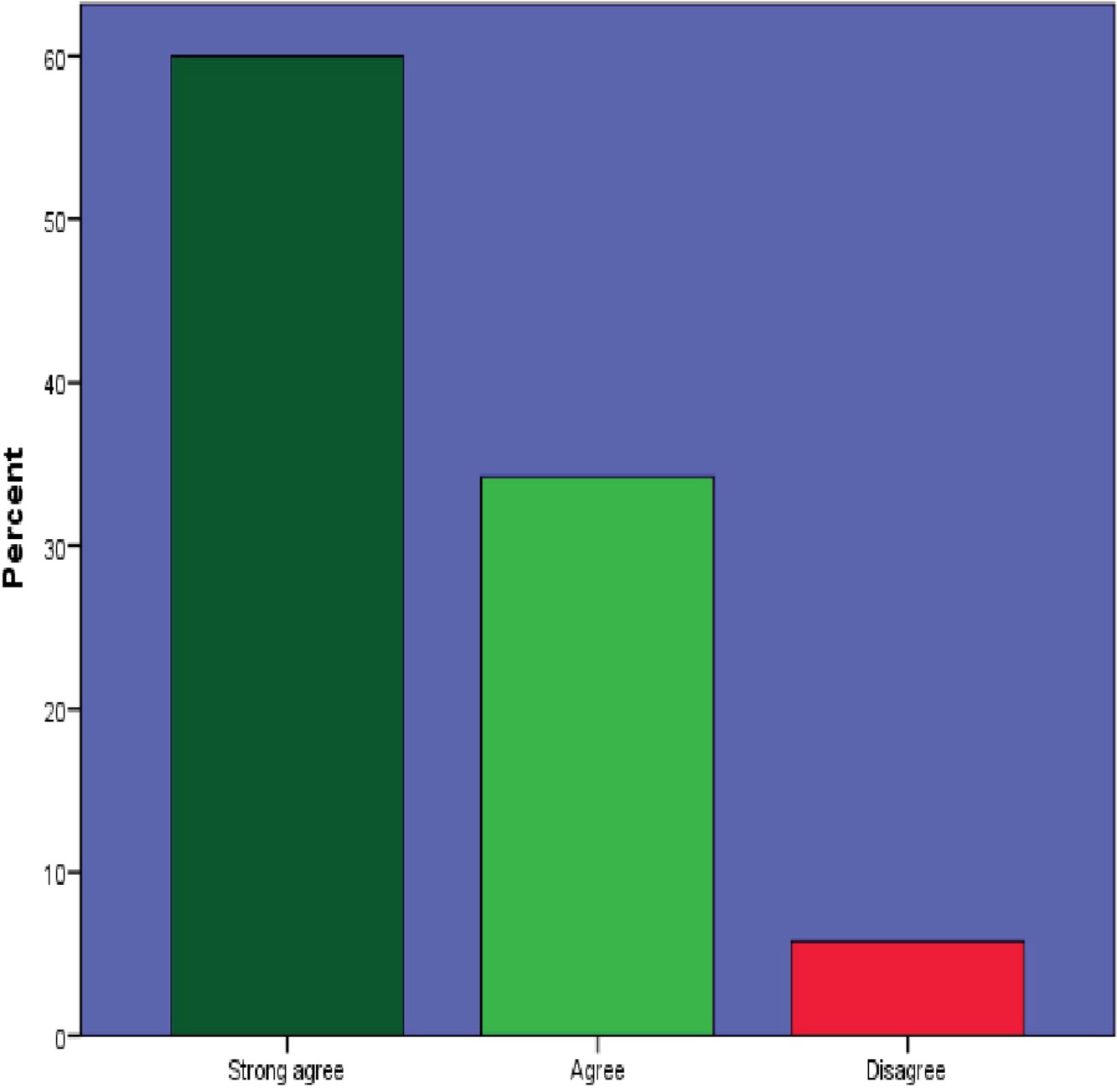

**Fig 3. Graphical representation of participants' willingness to utilize available healthcare services.**

or make them disappear entirely for health effects. The result of our study is similar to a study conducted by Ho-shut et al. (2015) in Taiwan [25].

Participants who were divorced at the time of this study utilized outpatient healthcare services twice as much as those who never married. This finding suggests that divorce can

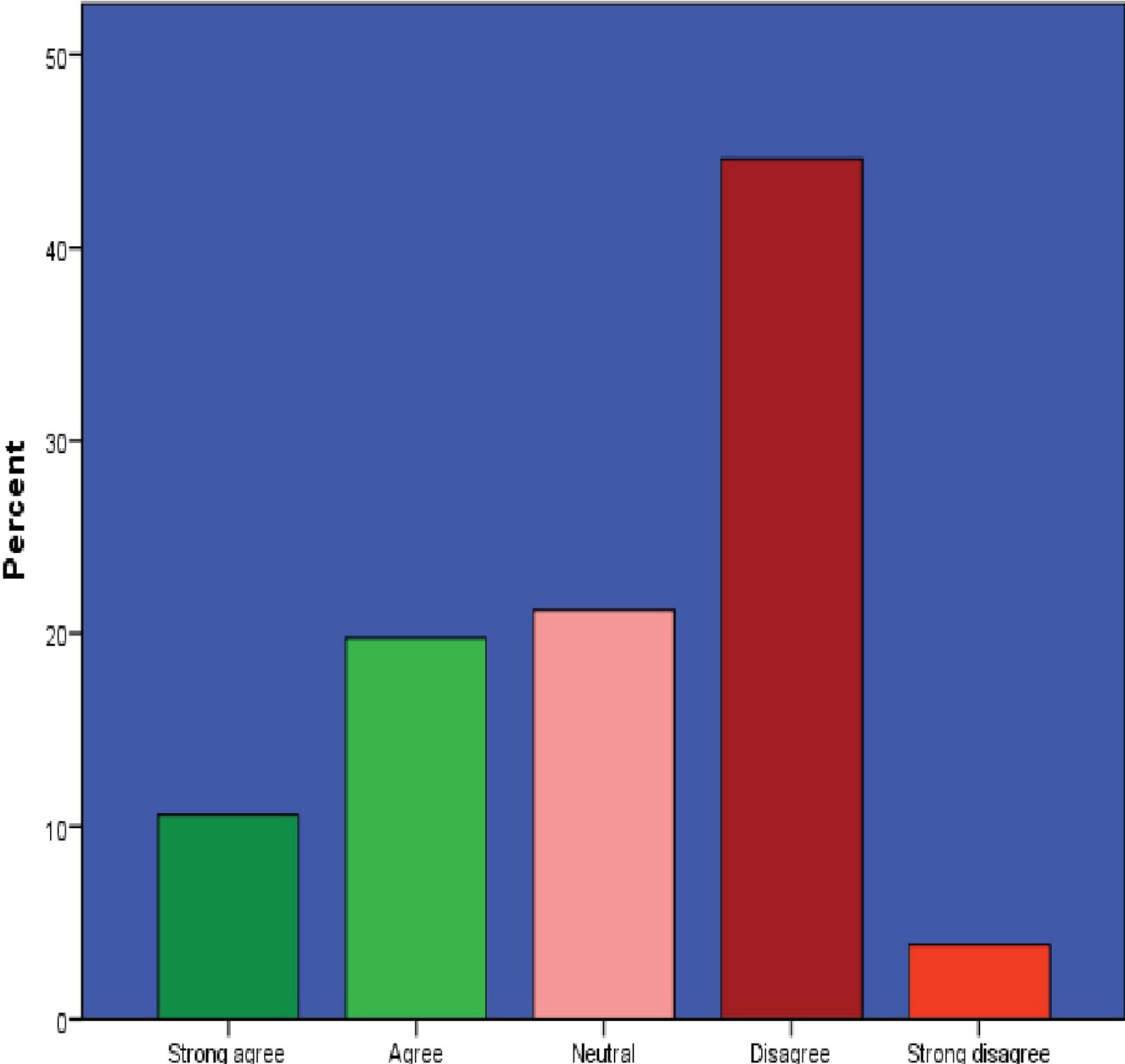

**Fig 4. Graphical depiction of participants' perception of susceptibility to diseases due to aging processes.**

significantly affect the health and well-being of individuals, leading to increased healthcare needs. Lin et al. (2019) reported that marital dissolution results in experiencing higher levels of depressive symptoms, resulting in increased and consistent healthcare utilization [26]. Healthcare providers must know this potential correlation and provide appropriate support and resources for divorced individuals to manage their health effectively [26].

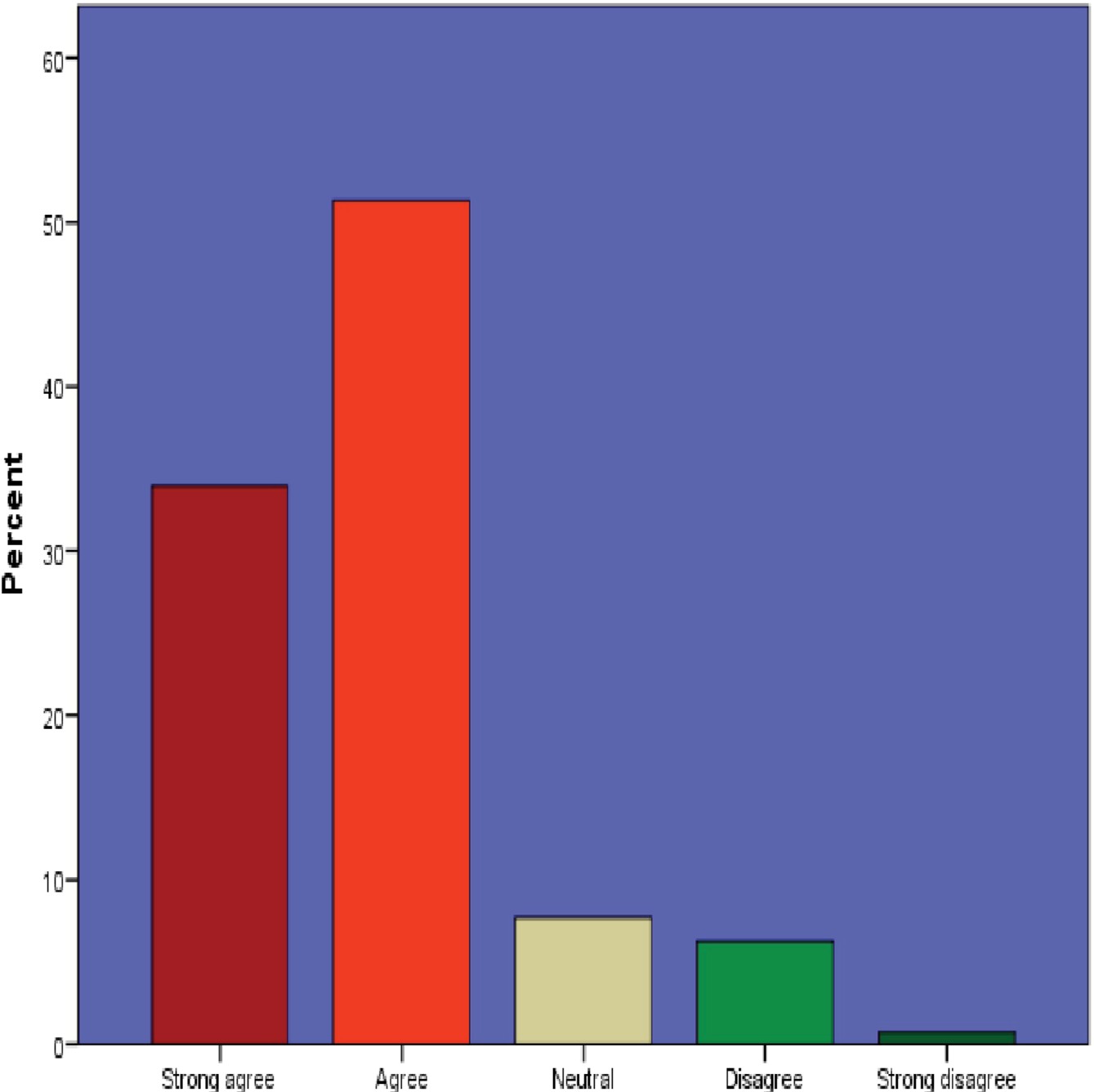

**Fig 5. Graphical depiction of participants' perception of the expensiveness of healthcare services.**

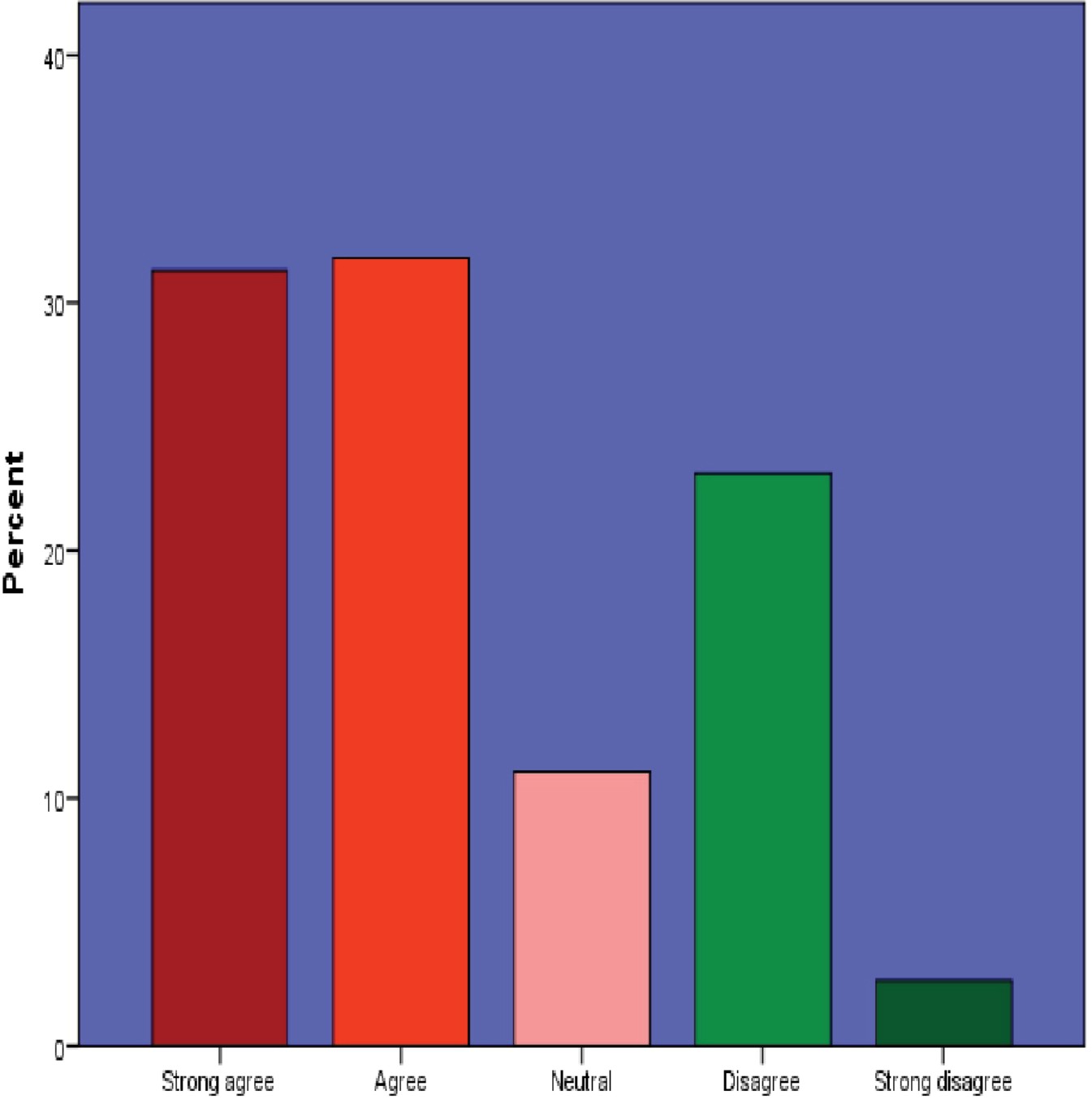

**Fig 6. Graphical representation of participants' perception of fear regarding disease diagnosis.**

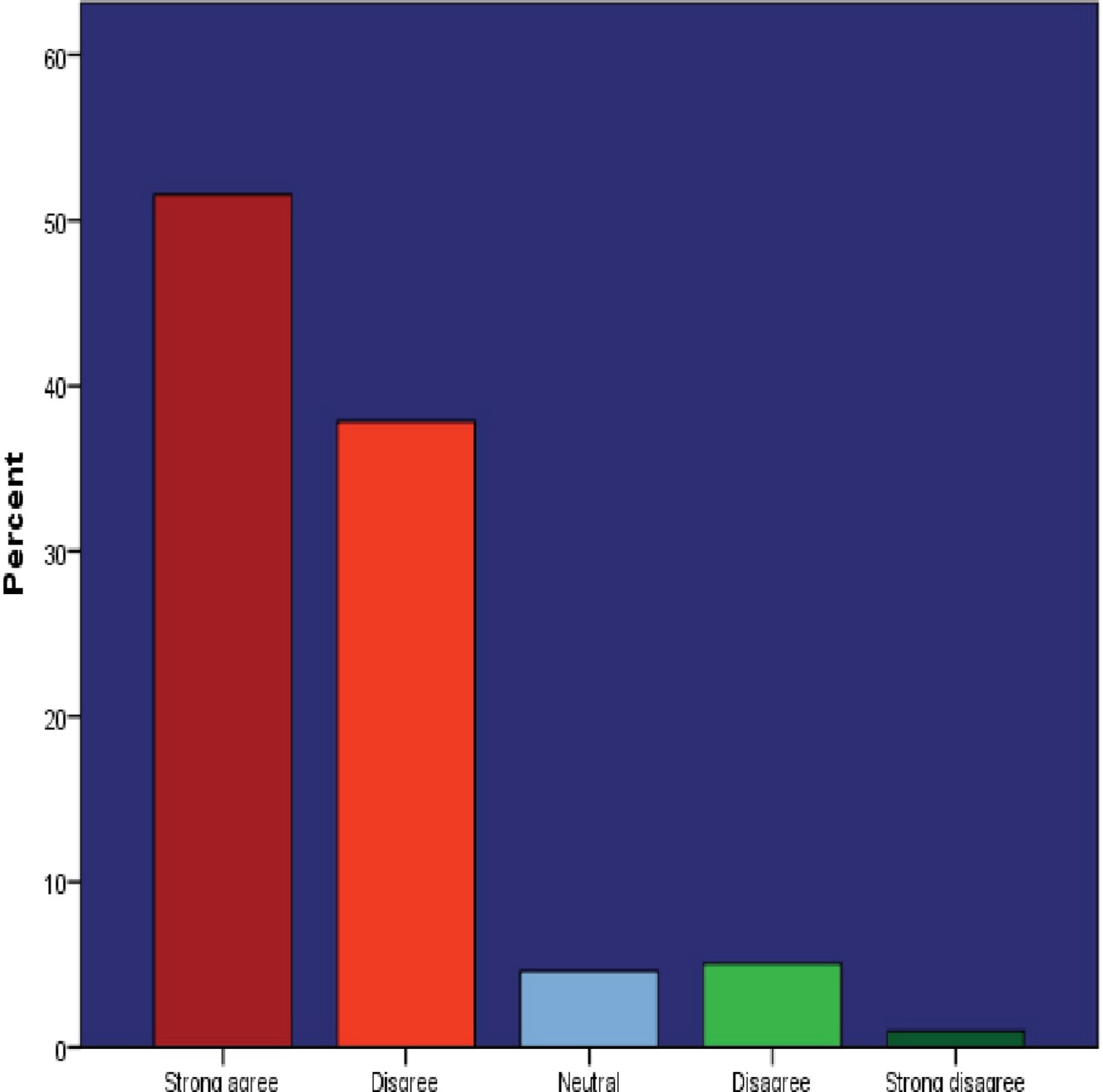

**Fig 7. Graphical depiction of participants' perception of health providers' rudeness during healthcare provision.**

In our study, we identified a significant correlation between individuals who maintained a predominantly favourable outlook on healthcare services and an increased likelihood of utilizing OPD services, with an odds ratio of 1.167, when compared to their counterparts. This discovery is in concordance with the results of another study conducted in Kibera, Nairobi, Kenya, in 2015, which reported analogous [21]. These findings substantiate the notion that contentment with healthcare provider services can wield considerable influence over individuals' perceptions of their access to healthcare and, in turn, their clinical outcomes [27]. The findings highlight the importance of individuals' satisfaction with healthcare provider services, indicating that being content in this aspect can greatly affect perceptions of healthcare accessibility and eventual clinical results. Thus, cultivating positive perceptions of healthcare services could play a vital role in promoting increased utilization of healthcare resources, ultimately contributing to enhanced health outcomes in communities.

Our study indicates a significant number of participants expressing concerns about being diagnosed with diseases. Similar trends are observed in other studies, including studies on nosophobia, the fear of severe illnesses, which has become more prominent, especially in the context of events like the COVID-19 pandemic [28,29]. Furthermore, a study on COVID-19-related fears illustrates variations influenced by factors such as gender, education level, and adherence to preventive measures, underscoring the complex interplay between fear and health viewpoints [30]. These collective findings highlight the widespread prevalence of health-related anxieties, reflecting broader patterns in healthcare perceptions and worries.

Our study revealed that a significant majority of study participants held the perception that healthcare providers might exhibit rudeness. This observation resonates with the findings of other relevant studies that have reported issues related to trust in healthcare. Studies have revealed that patients hold a perception that physicians are overwhelmed with heavy workloads, leading to hesitation in sharing their health concerns with them [31–33]. This finding is consistent with the broader understanding that the quality of services, patients' perceptions of healthcare providers, and their previous experiences with these providers collectively influence the utilization and acceptability of healthcare services [34]. Another study found that an elderly person's opinion of the doctor's lack of responsiveness was a larger disincentive to seeking care than more actual obstacles [35]. This underscores that alongside logistical challenges, the interpersonal dynamics between patients and healthcare providers significantly affect healthcare-seeking behaviours. Emphasizing patient-centred approaches in healthcare delivery, which prioritize trust, communication, and responsiveness alongside medical needs, is essential for ensuring equitable access and satisfaction. This complex relationship between service quality, patient perceptions, and past experiences shapes healthcare utilization and acceptability, making it imperative to address patient perceptions and experiences to enhance delivery and outcomes.

## Study limitations and mitigation

A cross-sectional study design had the advantage of being cost and time-efficient during data collection, as well as allowing researchers to compare numerous factors at the same time. However, the utilization of a cross-sectional design limits its ability to establish causal relationships. This limitation was mitigated by employing multivariate logistic regression to determine prevalence odds ratios. The absence of a longitudinal dimension further impedes the ability to draw causal inferences. To address this limitation, future research endeavours should consider incorporating longitudinal methodologies, enabling the exploration of temporal relationships and providing a more robust foundation for establishing causality. Moreover, the study employed convenient sampling to select the administrative region from thirty-one regions in Tanzania, potentially affecting the generalizability of the findings. Nevertheless, steps were

taken to mitigate this concern by employing a sizable sample size and adopting a multi-stage sampling approach for participant selection. These efforts were aimed at enhancing the representation of diverse perspectives within the studied population, thereby bolstering the study's validity and applicability.

## Conclusion

The study investigated the utilization of OPD healthcare services among the elderly in the Butiama and Musoma districts, revealing that 43.4% of participants accessed these services. Interestingly, individuals aged over 80 showed a significant decrease in OPD utilization. Additionally, widowed or divorced participants were found to be twice as likely to utilize OPD services, indicating the potential influence of widowhood or divorce on mental health and long-term care requirements. Further research is crucial to gain a deeper understanding of this context and offer customized assistance to divorced and widowed individuals.

This underutilization highlights the imperative for targeted interventions and strategies aimed at improving access and utilization, thereby enhancing healthcare outcomes for this demographic.

## Recommendations

In light of the research findings, stakeholders, policymakers, and healthcare providers must implement targeted interventions addressing the underutilization of OPD healthcare services among the elderly in the Butiama and Musoma districts of the Mara region. Prioritizing community outreach, education campaigns, and infrastructure improvements can effectively improve access. Acknowledging demographic variations, tailored support should be extended to those aged 80 years or older, ensuring their specific healthcare needs are met while recognizing and addressing increased utilization patterns among widowed and divorced/separated individuals. Healthcare providers must be aware of the impact of divorce on health, and implement support mechanisms like mental health services. Fostering positive patient-provider relationships, emphasizing effective communication and empathy, is vital for enhancing healthcare utilization. Acknowledging and working to improve perceptions of healthcare providers, and addressing concerns about rudeness, is crucial. Furthermore, additional research is recommended to identify and overcome context-specific barriers such as fear, shyness, and gender sensitivity, ultimately guiding the development of more effective interventions and improving healthcare outcomes for the elderly population. This research should concentrate on factors such as fear, shyness, and gender sensitivity.

## Supporting information

**S1 Questionnaire. Checklist guide for OPD utilization among elderly people.**
(DOCX)

**S1 File. Dataset of outpatient department utilization among elderly people in SPSS format.**
(SAV)

**S2 File. Dataset of outpatient department utilization among elderly people in excel format.**
(XLSX)

## Acknowledgments

We extend our sincere appreciation to MUHAS for their prompt and efficient issuance of ethical clearance. Furthermore, we express our profound gratitude to the District and Municipal

Medical Officers of Butiama and Musoma for generously providing the necessary permissions for conducting our research within their respective administrative jurisdictions.

## Author Contributions

**Conceptualization:** Magnus Michael Sichalwe, Chrisostom Charles Mwesiga.

**Data curation:** Magnus Michael Sichalwe, Chrisostom Charles Mwesiga.

**Formal analysis:** Magnus Michael Sichalwe, Chrisostom Charles Mwesiga.

**Funding acquisition:** Chrisostom Charles Mwesiga.

**Investigation:** Magnus Michael Sichalwe, Chrisostom Charles Mwesiga.

**Methodology:** Magnus Michael Sichalwe, Chrisostom Charles Mwesiga, Anna Tengia Kessy.

**Project administration:** Chrisostom Charles Mwesiga, Anna Tengia Kessy.

**Resources:** Chrisostom Charles Mwesiga.

**Supervision:** Anna Tengia Kessy.

**Validation:** Magnus Michael Sichalwe, Chrisostom Charles Mwesiga, Anna Tengia Kessy, Manas Ranjan Behera.

**Visualization:** Magnus Michael Sichalwe.

**Writing – original draft:** Magnus Michael Sichalwe, Chrisostom Charles Mwesiga, Manas Ranjan Behera.

**Writing – review & editing:** Magnus Michael Sichalwe, Chrisostom Charles Mwesiga, Anna Tengia Kessy, Manas Ranjan Behera.

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
