## [Editor Report · Decision Letter 0]

2 Jan 2024

PONE-D-23-36486Factors Affecting Utilization of Outpatient Healthcare Services among the Elderly Population in Butiama and Musoma Districts, Tanzania: A community-Based cross-sectional StudyPLOS ONE

Dear Dr. Mwesiga,

Thank you for submitting your manuscript to PLOS ONE. After careful consideration, we feel that it has merit but does not fully meet PLOS ONE’s publication criteria as it currently stands. Therefore, we invite you to submit a revised version of the manuscript that addresses the points raised during the review process.

**Dear Authors,**The manuscript is well designed and as an editor, I am interested in sending it to reviewers with a positive view for the peer review process. Although before doing so, I personally have the few queries (mentioned below in the additional comments section) that might make the study inferences more clear and would make the peer review process faster.==============================

We look forward to receiving your revised manuscript.

Kind regards,

Yogesh Kumar Jain, MPH

Academic Editor

PLOS ONE

Journal Requirements:

5. We note you have included a table to which you do not refer in the text of your manuscript. Please ensure that you refer to Table 4 in your text; if accepted, production will need this reference to link the reader to the Table.

Additional Editor Comments:

1. Please mention in the methods, how the wealth index was measured (any tool, or questionnaire based, if questionnaire based, which inputs were taken).

2. The recommendations section is more generic currently. Usually the existing policies and policy makers do have focus on the "super"-senior citizens (80 years and above), elderly are already aware about the importance of health utilisation and those who are aware, do themselves take more active OPD visits. Instead, I suggest to improve the recommendations based on your own inferences such as "fear, shyness, gender sensitivity", etc.

---

## [Author Response · Author response to Decision Letter 0]

7 Jan 2024

 Thank you for your valuable feedback. We have formatted the entire manuscript according to PLOS ONE's style requirements as requested.

 2. When completing the data availability statement of the submission form, you indicated that you would make your data available on acceptance. We strongly recommend all authors decide on a data-sharing plan before acceptance, as the process can be lengthy and hold up publication timelines. Please note that, though access restrictions are acceptable now, your entire data will need to be made freely accessible if your manuscript is accepted for publication. This policy applies to all data except where public deposition would breach compliance with the protocol approved by your research ethics board. If you are unable to adhere to our open data policy, please kindly revise your statement to explain your reasoning and we will seek the editor's input on an exemption. Please be assured that, once you have provided your new statement, the assessment of your exemption will not hold up the peer review process.

 Thank you for your valuable feedback. We acknowledge your request and we commit to provide the entire dataset upon the manuscript's acceptance.

 3. PLOS requires an ORCID iD for the corresponding author in Editorial Manager on papers submitted after December 6th, 2016. Please ensure that you have an ORCID iD and that it is validated in Editorial Manager.

 Thank you for your suggestion.

The ORCID iD for the corresponding author has already been supplied: it is 0009-0004-4332-3413.

 Thank you for your comment.

The title has already been revised in the manuscript to match the information provided in the online submission form.

 5. We note you have included a table to which you do not refer in the text of your manuscript. Please ensure that you refer to Table 4 in your text; if accepted, production will need this reference to link the reader to the Table. Thank you for bringing this to our attention.

We acknowledge the oversight and confirm that the necessary reference to Table 4 has been included in the revised manuscript. Page 12

 Thank you for your guidance.

Upon careful review, we confirm that each reference in our list is accurate and pertinent to the manuscript. We have ensured that there are no citations to retracted papers. No retracted articles are included in our reference list, and all citations align with relevant sources. There are no changes required in this regard for the revised manuscript.

In case of any specific concerns or further clarifications needed, please do not hesitate to let us know.

 Additional Editor Comments: 

 1. Please mention in the methods, how the wealth index was measured (any tool, or questionnaire-based, if questionnaire-based, which inputs were taken). 

Thank you for your feedback.

The measurement of the wealth index has been addressed in the methods section of the manuscript. We have shared information about the utilized tool, which is questionnaire-based. Please refer to the supplementary file for details on Question 12 of the questionnaire. 

The relevant inputs for the wealth index calculation have also been specified. If you have any additional questions or if further clarification is needed, please feel free to let us know. Available in manuscript page number 5 

 2. The recommendations section is more generic currently. Usually, the existing policies and policymakers do have a focus on the "super"-senior citizens (80 years and above), the elderly are already aware of the importance of health utilisation and those who are aware, do themselves take more active OPD visits. Instead, I suggest improving the recommendations based on your inferences such as "fear, shyness, gender sensitivity", etc. 

Thank you for your insightful suggestion.

We appreciate your feedback on the recommendations section. The manuscript has been revised to enhance the specificity of recommendations. We believe these adjustments provide a more nuanced and context-specific set of recommendations. If you have any further recommendations or specific areas of concern, please feel free to let us know. Page number 16 of manuscript.

---

## [Decision Letter · Decision Letter 1]

13 Mar 2024

PONE-D-23-36486R1Factors Affecting Utilization of Outpatient Healthcare Services among the Elderly Population in Butiama and Musoma Districts, Tanzania: A community-Based cross-sectional StudyPLOS ONE

Dear Dr. Mwesiga,

Thank you for submitting your manuscript to PLOS ONE. After careful consideration, we feel that it has merit but does not fully meet PLOS ONE’s publication criteria as it currently stands. Therefore, we invite you to submit a revised version of the manuscript that addresses the points raised during the review process.

Manuscript has been well constructed with good statistical analysis and received a positive reviewer feedback. Please improve as per the detailed suggestions provided by the reviewers.

We look forward to receiving your revised manuscript.

Kind regards,

Yogesh Kumar Jain, MPH

Academic Editor

PLOS ONE

Additional Editor Comments:

The manuscript has received positive feedback from the reviewers and has been provided detailed suggestions for improvement. Kindly refine accordingly and resubmit.

Reviewers' comments:

Reviewer's Responses to Questions

**Comments to the Author**

1. If the authors have adequately addressed your comments raised in a previous round of review and you feel that this manuscript is now acceptable for publication, you may indicate that here to bypass the “Comments to the Author” section, enter your conflict of interest statement in the “Confidential to Editor” section, and submit your "Accept" recommendation.

Reviewer #1: (No Response)

2. Is the manuscript technically sound, and do the data support the conclusions?

Reviewer #1: Partly

3. Has the statistical analysis been performed appropriately and rigorously? 

Reviewer #1: Yes

4. Have the authors made all data underlying the findings in their manuscript fully available?

Reviewer #1: Yes

5. Is the manuscript presented in an intelligible fashion and written in standard English?

Reviewer #1: Yes

6. Review Comments to the Author

Reviewer #1: Page 2 : “Longer life expectancies offer societal benefits but also bring complex challenges tied to an

ageing population, impacting productivity, economic growth, and social security sustainability”. I would briefly specify some societal benefits and , as you state in following paragraphs, I would specify those aspects of aging that have a negative impact including t morbidity and potential reduction of quality of life . This would contextualize better the need to contrast those negative aspects of aging by increasing care, in order to benefit more of the listed benefits of longer life expectancies. Could this do?

Page 3 study design : “An advantage of a cross-sectional study design was its convenience in terms of cost and time in data collection, as well as allowing the researcher to compare multiple variables simultaneously.” This paragraph could be moved as a “strength of the study” in the limits section.

Page 3: study setting : could we specify better the setting ? for example, is it a rural or urbanised setting ? what is the main activity of the population in general, etc ? This would make us contextualise better the results….

Page 4: the districts were selected by convenience : could you briefly specify what made their selection convenient ?

Page 5 : independent variables : I would move the following paragraph to the statistics analysis section, because it specifies how the collected variables are analysed : “To calculate the weighted household wealth index, households were scored based on their possession of consumer goods or durable assets. These scores were determined through principal component analysis (PCA) in SPSS, which reduced the variables into components using loading factors as weights, ultimately yielding the weighted wealth index. Following this, households were categorized into four equal wealth quartiles, and the scores were presented as lowest, Second, Middle, and Highest wealth index”.

Page 5 : data collection methods: you refer to research instruments, including a questionnaire, but there are no details on the questionnaire : how was it structured, the language used ( English, Kiswahili etc , how was it sub ministered and by who ….

Page 6 : in addition to the percentage , specify the ratio : for example x widows/ total participants

Table 2 : could we specify in the Table the p value set for significant results? Could we also add a star on those values in table that meet significance , on the p value column ?

Table 3 : title doesn’t match the content : it contains not only socio-economic factors, but also anagraphic factors, and risk factors like smoke and alcohol intake . put a star on significant values

Table 4 : could data be presented as a graph? It could provide an easier way of perceiving the frequencies of answers in a rapid visual assessment of data

Discussion : I would move the following sentence in the last part of discussion/ conclusions “This underutilization highlights the imperative for targeted interventions and strategies aimed at improving access and utilization, thereby enhancing healthcare outcomes for this demographic.”

Page 14 : check direction of the association for the variables in this sentence : “In this study, being older than 80 years was associated with declining utilization of OPD’s healthcare services among elders “ ….it should be the contrary, over eighty they seem to use more…..it is the 70-80 which underutilises it.

The discussion is too short, and it is to narrow when it comes to the discussion of table number 4, which gives some inputs for expansion….

7. PLOS authors have the option to publish the peer review history of their article (what does this mean?). If published, this will include your full peer review and any attached files.

Reviewer #1: No

---

## [Author Response · Author response to Decision Letter 1]

29 Mar 2024

Dear Reviewer,

I hope this message finds you well. I am writing to express my sincere gratitude for taking the time to review our manuscript titled “Factors affecting utilization of outpatient healthcare services among the elderly population in Butiama and Musoma districts, Tanzania: A community-based cross-sectional study" submitted to PLOS ONE. Your insightful comments and constructive feedback have been invaluable in enhancing the quality and clarity of our research.

We have carefully reviewed all the points raised in your review and have made revisions accordingly to address them. While comprehensive responses are provided in both the rebuttal letter and the revised manuscript, I would like to provide you with a concise summary of the changes made in response to your valuable feedback:

1. Societal Benefits and Negative Impacts of Aging: We have expanded the discussion to include specific examples of societal benefits and negative impacts of ageing to provide a better context for the study.

2. Study Design and Setting: The paragraph highlighting the advantages of the study design has been moved to the strengths section, and additional details about the study setting, including rural or urban classification and main activities of the population, have been incorporated.

3. Convenience Sampling: We have clarified the factors that made the selection of districts convenient in the methodology section.

4. Independent Variables and Data Collection Methods: The paragraph describing the analysis of independent variables has been moved to the statistics analysis section, and more details about the questionnaire, including its structure, language, and administration, have been provided.

5. Additional Details for Clarity: Ratios in addition to percentages have been included, the significance level in tables has been specified, stars have been added to indicate significant values in tables, and consideration has been given to presenting data from Table 4 in a graphical format.

6. Discussion: The suggested sentence has been moved to the last part of the discussion/conclusion section, and the discussion has been expanded to provide further insights based on the findings presented in Table 4.

7. Correction in Interpretation: The interpretation regarding the association between age groups and utilization of OPD services has been corrected.

We believe that these revisions have strengthened the manuscript and improved its overall quality.

Once again, I extend my sincere appreciation for your thorough review and valuable suggestions. Your expertise and guidance have been instrumental in refining our work. We remain committed to upholding the highest standards of scientific integrity and excellence.

Thank you for your continued support and consideration.

Warm regards,

Sichalwe M.

---

## [Editor Report · Decision Letter 2]

16 May 2024

Factors Affecting Utilization of Outpatient Healthcare Services among the Elderly Population in Butiama and Musoma Districts, Tanzania: A community-Based cross-sectional Study

PONE-D-23-36486R2

Dear Dr. Mwesiga,

We’re pleased to inform you that your manuscript has been judged scientifically suitable for publication and will be formally accepted for publication once it meets all outstanding technical requirements.

Kind regards,

Yogesh Kumar Jain, MPH

Academic Editor

PLOS ONE
---

## [Editor Report · Acceptance letter]

12 Jun 2024

PONE-D-23-36486R2 

PLOS ONE

Dear Dr. Mwesiga, 

I'm pleased to inform you that your manuscript has been deemed suitable for publication in PLOS ONE. Congratulations! Your manuscript is now being handed over to our production team.

Kind regards, 

on behalf of

Dr. Yogesh Kumar Jain 

Academic Editor

PLOS ONE